# Simple Rate Expression for Catalyzed Ammonia Decomposition for Fuel Cells

**DOI:** 10.3390/molecules28166006

**Published:** 2023-08-10

**Authors:** Robert B. Barat

**Affiliations:** Otto H. York Department of Chemical and Materials Engineering, New Jersey Institute of Technology, Newark, NJ 07102, USA; robert.b.barat@njit.edu

**Keywords:** catalyst, ammonia, mechanism, decomposition, CSTR, packed bed

## Abstract

This paper examines NH_3_ decomposition rates based on a literature-proven six-step elementary catalytic (Ni-BaZrO_3_) mechanism valid for 1 × 10^5^ Pa pressure in a 650–950 K range. The rates are generated using a hypothetical continuous stirred tank catalytic reactor model running the literature mechanism. Excellent correlations are then obtained by fitting these rates to a simple overall kinetic expression based on an assumed slow step, with the remaining steps in fast pseudo-equilibria. The robust overall simple rate expression is then successfully demonstrated in various packed bed reactor applications. This expression facilitates engineering calculations without the need for a complex, detailed mechanism solver package. The methodology used in this work is independent of the choice of catalyst. It relies on the availability of a previously published and validated elementary reaction mechanism.

## 1. Background

Motivated by global warming, the worldwide movement to decarbonize national economies continues to grow. Fuels high in hydrogen content (e.g., CH_4_ or pure H_2_) are preferred. Compressed natural gas for transportation fleets is well known. The popularity and performance of battery-powered pure electric vehicles are growing despite limited charging stations and relatively long battery recharging times.

As a non-carbon fuel, liquid NH_3_ is attracting increasing attention [1]. It combines the advantages of easy storage and transport without producing greenhouse gases. Anhydrous liquid NH_3_ is typically stored at 9.6 × 10^5^ Pa absolute at room temperature. “Green” NH_3_ can be catalytically produced from N_2_ and H_2_ using solar power [2]. Ammonia as a H_2_ source is gaining considerable attention [3].

While direct NH_3_ combustion produces H_2_O and N_2_ as major products, temperatures can easily exceed 1000 K, resulting in pollutant nitrogen oxides [4]. To avoid these temperatures, fuel cells are preferred for the oxidation of the H_2_ from NH_3_ decomposition [2,5]. Ammonia-powered fuel cells are now being tested for freight-hauling vehicles, including large trucks [6] and locomotives [7].

Production of H_2_ from NH_3_ can be accomplished in various ways, including catalytic decomposition from either gaseous or solution NH_3_ [3]. Various metals (e.g., Ru, Ir, Ni, and Rh) and supports have been investigated as catalysts for the decomposition of gaseous NH_3_ [3,8]. While Ru shows excellent activity, its high cost and limited availability make Ni more attractive [8]. An alternative approach is the decomposition of NH_3_ in solution using electrocatalysts [3].

In the typical ammonia-powered fuel cell, H_2_ produced from gaseous NH_3_ decomposition yields protons at the anode, thus releasing electrons into the external load circuit. The protons diffuse across a membrane to the cathode, where they react with O_2_ (typically from air) to form H_2_O vapor, and the electron circuit is completed. In ceramic fuel cells, temperatures can exceed 1000 K due to H_2_ oxidation exothermicity. At this level, the kinetics of the endothermic NH_3_ decomposition are sufficiently fast that NH_3_ can directly feed the fuel cell [9]. However, the risk of NO_x_ formation still exists. Therefore, care is needed for NH_3_ decomposition catalyst and fuel cell designs to avoid these temperatures.

In this paper, a simple kinetic *engineering* model is developed for NH_3_ decomposition over a Ni-BaZrO_3_ catalyst. This model is calibrated over a relevant temperature range using reaction rates obtained from a detailed published elementary reaction mechanism [9]. The performance of this model is then compared to that of the detailed mechanism in simulations of packed bed (plug flow PBR) ideal adiabatic reactors. The utility of the engineering model is further demonstrated with simulations of PBRs with heat transfer and H_2_ diffusion. This simple model is intended to facilitate quick engineering calculations and screening studies for fuel cell design.

It should be noted that the purpose of this paper is the demonstration of a calculation methodology assuming the availability of a proven, detailed reaction mechanism. Catalyst choice, design, and effectiveness are beyond the scope of this work.

## 2. Equilibrium

Ammonia decomposition [$NH_{3}\to 0.5N_{2}+{1.5H}_{2}$] is endothermic (${∆H}_{298K}^{o}$
*=* 45,900 J) and becomes thermodynamically favorable above approximately 450 K (at 1 × 10^5^ Pa). Figure 1 shows the equilibrium conversion over a wide temperature range at 1 × 10^5^ Pa pressure for pure NH_3_ feed, with complete conversion above 650 K. The calculation uses an online equilibrium calculator [10], with results consistent with published species thermodynamic properties [11]. The temperature range of this study is 650–950 K at 1 × 10^5^ Pa pressure. Therefore, the decomposition is not thermodynamically limited under the conditions of this study.

## 3. Kinetics and Mechanism

A common mechanism in the literature [9] for NH_3_ decomposition on a Ni-based catalyst consists of the following six *elementary* reversible steps:$$\;1.\;NH_{3}+S=NH_{3}\cdot S$$
$$\;2.\;NH_{3}\cdot S+S=NH_{2}\cdot S+H\cdot S$$
$$\;3.\;NH_{2}\cdot S+S=NH\cdot S+H\cdot S$$
4. NH·S+S=N·S+H·S
$$5.\;N\cdot S+N\cdot S=N_{2}+2S$$
$$6.\;H\cdot S+H\cdot S=H_{2}+2S$$
where *S* represents an unoccupied catalytic site, and #·S is an adsorbed species. Zhu et al. [9] validated this mechanism for a Ni-BaZrO_3_ catalyst over ~650–950 K range at 1 × 10^5^ Pa pressure using experimental data from Okura et al. [8]. The kinetic parameters are presented there for both the forward and reverse steps. This eliminates the need for any estimates of thermodynamics for surface-adsorbed species that would be needed if kinetic parameters were provided for only the forward steps.

The entire mechanism is “thermodynamically consistent”. This means that accurate (thermodynamically) reaction equilibrium gas compositions (NH_3_, N_2_, and H_2_) are achieved if a time-dependent (kinetic) ammonia decomposition PBR flow reactor simulation is carried out for a sufficiently long time. These kinetic parameters are used in this study as described below. This was demonstrated by Karakaya et al. [12] with an elementary catalytic mechanism for methane oxidative coupling.

A two-pronged strategy is used in the current study. In the first portion, simulations of catalyzed NH_3_ decomposition are performed using continuously stirred tank reactor (CSTR, perfectly mixed) calculations together with the detailed mechanism shown above. These provide rates of NH_3_ decomposition as functions of NH_3_, H_2_, and N_2_ partial pressures. These rates are then used to calibrate a proposed single overall kinetic expression for NH_3_ decomposition derived from a fast pseudo-equilibrium analysis applied to the detailed six-step mechanism shown above. In the second portion, the calibrated single kinetic rate expression is used in simple packed bed reactor (PBR, perfect plug flow) simulations to demonstrate its utility in engineering and screening calculations.

## 4. Simplified Overall Rate Expression

The six-step, detailed elementary reaction mechanism described above is authoritative for kinetic reactor simulations. However, such simulations require a computation package such as *Detchem*^®^ [13]. It is often desirable to reduce such mechanisms to a simple kinetic rate expression for relatively quick engineering calculations. Such a reduction is applied here using the fast pseudo-equilibrium approach based on the Langmuir–Hinshelwood algorithm [14].

### 4.1. Derivation by Langmuir–Hinshelwood Algorithm

The algorithm applied to the six-step mechanism begins with choosing a rate-determining (slow) step. The slow step is taken as effectively irreversible. The remaining steps are assumed to be in fast pseudo-equilibrium (FPE). Consider Step 5 to be the slow step, as recommended by Bell and Torrente-Murciano [15] for a Ni-based NH_3_ decomposition catalyst:(1)$$-r_{NH3}^{”}=r_{slow}\equiv r_{5}=k_{5}C_{N\cdot S}^{2}$$
where $C_{j\cdot S}$ is the surface concentration of the adsorbed species j, and C_S_ is the concentration of vacant sites. Step 4 can be used to estimate $C_{N\cdot S}$ by applying the FPE:(2)$$r_{4}=k_{4}C_{NH\cdot S}C_{S}-k_{-4}C_{N\cdot S}C_{H\cdot S}=k_{4}\left[C_{NH\cdot S}C_{S}-C_{N\cdot S}C_{H\cdot S}/K_{4}\right]$$
where $K_{4}=k_{4}/k_{-4}$. The FPE claims $r_{4}/k_{4}\approx 0$, which implies:(3)$$C_{N\cdot S}\approx C_{NH\cdot S}C_{S}K_{4}/C_{H\cdot S}$$

Similarly, steps 3, 2, 1, and 6 are used in succession to derive $C_{NH\cdot S}$, $C_{NH2\cdot S}$, $C_{NH3\cdot S}$, and $C_{H\cdot S}$, respectively. Substitution as needed produces a preliminary rate expression:(4)$$-r_{NH3}^{”}=k_{5}{\hat{K}}^{2}C_{S}^{2}P_{NH3}^{2}/P_{H2}^{3}$$

The *C_S_* derives from the “site balance”: $C_{T}=C_{S}+C_{NH3\cdot S}+C_{NH2\cdot S}+C_{NH\cdot S}+C_{N\cdot S}+C_{H\cdot S}$. Bell and Torrente-Murciano [15] suggest that adsorbed N atoms are the dominant adsorbed species. Therefore, the site balance simplifies to: $C_{T}\approx C_{S}+C_{N\cdot S}$, from which:(5)$$C_{S}\approx C_{T}/\left(1+P_{NH3}\hat{K}/P_{H2}^{1.5}\right)$$

Substituting Equation (5) into (4) reveals the overall rate expression, assuming desorption and recombination of N atoms (Step 5) is rate-determining:(6)$$-r_{NH3}^{”}=kP_{NH3}^{2}/{\left(P_{H2}^{1.5}+{\hat{K}P}_{NH3}\right)}^{2}$$
where k and $\hat{K}$ are lumped constants: $k\equiv k_{5}{\hat{K}}^{2}C_{T}^{2}$ and $\hat{K}\equiv K_{1}K_{2}K_{3}K_{4}K_{6}^{3}$. For the remaining discussion below, the parameter $\hat{k}$ replaces $\hat{K}$.

### 4.2. Preparation for Calibration of Overall Rate Expression

The CSTR calculation offers a simple way to generate rate data. The ideal CSTR model assumes perfect mixing, with the rate obtained from the steady CSTR species balance. For the NH_3_ decomposition:(7)$$r_{NH3}^{”}=F_{NH3,o}X_{NH3}/A_{CSTR}$$

In this study, the species net rates are related by the stoichiometry: $NH_{3}\to 0.5N_{2}+{1.5H}_{2}$.
(8)$$-r_{NH3}^{”}=r_{N2}^{”}/0.5=r_{H2}^{”}/1.5$$

Using a familiar nomenclature (Fogler, 2020), the species partial pressures are given by:(9)$$P_{j}=P_{NH3,o}\left(\theta_{j}+\nu_{j}X_{NH3}\right)/\left(1+\epsilon X_{NH3}\right)$$
where $\theta_{j}\equiv F_{j,o}/F_{NH3,o}$ and $\epsilon \equiv y_{NH3,o}\delta$. For the above overall stoichiometry, δ=1. The stoichiometric coefficients are: $\nu_{NH3}=-1$, $\nu_{N2}=0.5$, $\nu_{H2}=1.5$.

The lumped parameters in the simplified rate expression (Equation (6)) will be calibrated using rate data obtained from *Detchem*^®^ CSTR^®^ simulations run with the detailed NH_3_ decomposition mechanism [8] shown above. Available online [13] with numerous applications, the *Detchem*^®^ software package is widely used [16,17] and quite versatile. This package executes detailed material, energy, and momentum balances using a user-supplied elementary reaction mechanism and required reactor input and parameter information.

Since the detailed mechanism is validated on experimental data [8], using the simulated rate data here obtained with this mechanism is equivalent to calibrating against experimental data directly. Because the Zhu et al. [8] detailed mechanism does such an excellent job modeling the X_NH3_ vs. temperature data of Okura et al. [8], calculations with it can be used as the “data” source against which the overall rate expression (Equation (6)) is calibrated. The same catalyst, temperature, and pressure range as Okura et al. [8] are used.

In this study, separate packed bed (plug flow) and continuous stirred tank (perfectly mixed) reactor simulations were performed with *Detchem*^®^ PBED^®^ and CSTR^®^ applications, respectively. The governing PBED^®^ and CSTR^®^ equations are described in the *Detchem*^®^ manual [13] and listed elsewhere [18].

At a given temperature and pressure, the *Detchem*^®^ CSTR^®^ application is run over a wide range of pure NH_3_ feed rates *F_NH3,o_* to generate gaseous reactor effluent mole fractions, y_j_. Any pressure drop across the reactor is assumed to be small enough to ignore. The exit mole fractions are used to calculate the corresponding partial pressures, P_j_. Knowing P_j_, Equation (9) is used to determine the corresponding NH_3_ conversion, *X_NH_*_3_:(10)$$P_{NH3}=P_{NH3,o}\left(1-X_{NH3}\right)/\left(1+X_{NH3}\right)$$

Similarly, *P_H2_* and *P_N2_* are used to test for *X_NH3_* consistency:(11)$$P_{H2}=P_{NH3,o}\left(1.5X_{NH3}\right)/\left(1+X_{NH3}\right)\;\;\;\;\;P_{N2}=P_{NH3,o}\left(0.5X_{NH3}\right)/\left(1+X_{NH3}\right)$$

With X_NH3_ data in hand, Equation (7) is used to generate rate $r_{NH3}^{”}$ data. At the given temperature, the partial pressure, conversion, and rate $\left(P_{j}{,\;X}_{NH3},r_{NH3}^{”}\right)$ data sets, based on exit mole fraction data from the *Detchem*^®^ CSTR^®^ runs and Equations (7), (10), and (11), are used below to calibrate the overall rate expression (Equation (6)) through the lumped constants k and $\hat{k}$.

## 5. Calibration of the Simplified Overall Rate Expression

An extensive series of *Detchem*^®^ CSTR^®^ simulations with a pure NH_3_ feed over a wide temperature range was performed. Details appear in Table 1. The runs were isothermal at constant pressure. The feed rate range for each temperature was chosen such that the X_NH3_ ranged from near zero to almost 1. The gas volume and catalytic surface area values listed are consistent with the PBED^®^ simulations described later.

For a given temperature and pure NH_3_ feed, all $\left(P_{j}{,\;X}_{NH3},r_{NH3}^{”}\right)$ data sets were used to calibrate the Equation (6) overall rate expression through the constants k and $\hat{k}$. It should be noted that the Langmuir–Hinshelwood algorithm was used to derive different overall rate expressions, such as Equation (6), each based on a different assumed slow step. All expressions were subjected to testing with the generated $\left(P_{j}{,\;X}_{NH3},r_{NH3}^{”}\right)$ data sets. These results are not shown here because the only statistically acceptable rate form was Equation (6), which is based on an assumed slow step of N adatom recombination, which is consistent with Bell and Torrente-Murciano [15].

### 5.1. Calibration Results

The generated $\left(P_{j}{,\;X}_{NH3},r_{NH3}^{”}\right)$ data sets show that, for all relevant temperatures, the rates decrease with increasing NH_3_ conversion. Sample results are shown in Figure 2 (650 K) and Figure 3 (950 K). The simple overall rate expression (Equation (6)) does an excellent job for all pure NH_3_ feed cases. It is satisfying that the overall rate expression shows the inflections in the curves at higher X_NH3s_.

With excellent modeling obtained at all five temperatures for the pure NH_3_ feed, the rate parameters *k* and $\hat{k}$ as functions of temperature are presented in Figure 4. The non-Arrhenius curves are each regressed to the form:(12)$$ln\left(k\;or\;\hat{k}\right)=a+b\left(1/T\right)+c{\left(1/T\right)}^{2}$$

The values of a, b, and c are presented in Table 2. This form is for the sake of convenience. The modest non-linearities in Figure 4 do not necessarily mean a change in rate-determining step over the temperature range. Both parameters k and $\hat{k}$ are *lumped*, consisting of multiple single constants, each with its own temperature dependence. For example, $\hat{k}$ reflects both dissociative adsorptions and an associative desorption, which would have contrary temperature dependencies.

Figure 2 and Figure 3 present additional curves, based on the *calibrated* Equation (6), and data points, based on *Detchem*^®^ CSTR^®^ simulations, wherein 25% of the NH_3_ feed is replaced with H_2_ or N_2_ at the same total feed rate. The H_2_ cases result in lower NH_3_ decomposition rates, as suggested by Equation (6). The impact is more pronounced at lower temperatures. The impact of adding N_2_ in this role is less clear. Equation (6) suggests little impact beyond a dilution effect on H_2_ and NH_3_ partial pressures. In both of these tests, the overall simplified rate expression (Equation (6)) calibrated on pure NH_3_ feed runs did well in predicting the detailed mechanism-based rate results.

## 6. PBR Species and Energy Balances Used with Overall Rate Expression

The overall calibrated rate expression is now used to simulate NH_3_ decomposition in the packed bed reactor found in an idealized ammonia fuel cell. The species balances used are shown in Equations (13) and (14). The PBR assumes perfect plug flow.
(13)$$dF_{NH3}/dz={r_{NH3}^{”}a}_{v}A_{c}\;dF_{N2}/dz={r_{N2}^{”}a}_{v}A_{c}$$
(14)$$dF_{H2}/dz={r_{H2}^{”}a}_{v}A_{c}-k_{c}P_{H2}\;$$

Zhu et al. [9] show that the NH_3_ decomposition can occur within the anode structure of the fuel cell or in the catalytic zone outside the fuel cell itself. If inside, the “diffusion” term on the right side of Equation (14) crudely approximates the passage of H_2_ through the fuel cell membrane to the cathode, where it is oxidized. If outside, then k_c_ = 0. The PBR energy balance is:(15)$$\frac{dT}{dZ}=\frac{\dot{q}+r_{NH3}^{”}∆H_{r,NH3}a_{v}A_{c}}{F_{NH3}c_{p,NH3}+F_{N2}c_{p,N2}+F_{H2}c_{p,H2}}$$

If the PBR is assumed adiabatic, then $\dot{q}=0$. If heat from the cathodic oxidation of H_2_ conducts through the fuel cell membrane to the anode to satisfy some of the NH_3_ decomposition endothermicity, the heat transfer rate can be crudely approximated by:(16)$$\dot{q}=fk_{c}P_{H2}\left(-∆H_{r,H2O}\right)$$

The thermal data (*c_p,j_* and $\Delta H_{r,j})$ are presented in the Appendix A. Constant pressure and ideal gas are assumed throughout this study. Equations (17) provide additional values: (17)$$X_{NH3}=\left(F_{NH3,o}-F_{NH3}\right)/F_{NH3,o}\;\;\;\;\;P_{j}=\left(F_{j}/F_{T}\right)P\;\;\;\;\;F_{T}=\sum_{j}F_{j}$$

### 6.1. Packed Bed Reactor Simulations

We begin with packed bed calculations performed using the *Detchem*^®^ PBED^®^ package. Table 3 presents the data for the simulations. Both high and low pure NH_3_ feed rates were used, together with a range of feed temperatures. These runs produce the data against which the single rate expression engineering model will be tested.

The engineering model (Equations (13)–(17)) uses the overall single rate expression (Equation (6)) together with the temperature-fitted rate constants in Equation (12). The *Polymath*^®^ ODE package [19] solved the engineering model for the packed bed reactor.

Figure 5 and Figure 6 present the comparative results. Results in Figure 5 for a pure NH_3_ feed (1 × 10^−5^ mole/s) at 800 K show that the engineering model does an excellent job predicting the detailed mechanism axial profiles for both NH_3_ content and reactor temperature. Similar excellent results are shown in Figure 6 over a range of feed temperatures at a higher feed rate (1 × 10^−4^ mol/s). The fits of the NH_3_ mole fractions are excellent, while the temperatures are predicted within 3 K. With the credibility of the engineering model with single overall rate expression now established, this model was applied for varied applications for which *Detchem*^®^ PBED^®^ is not applicable.
molecules-28-06006-t003_Table 3Table 3Data for *Detchem*^®^ PBED^®^ and Engineering Model PBR Simulations.*Bed radius:* 5 × 10^−3^ m*Bed length:* 0.1 m *Catalyst area/bed volume:* 2.19 × 10^5^ m^−1^
*Total catalyst area:* 0.172 m^2^
*Bed porosity:* 0.38 *Particle diameter:* 2 × 10^−5^ m *Feed:* pure NH_3_
*Total feed rates:* 1 × 10^−5^, 1 × 10^−4^ mole/s *Pressure:* 1 × 10^5^ Pa (constant)*Feed temperature range:* 650–950 K *Diffusion coefficient k_c_ (Equation (5)):* 5 × 10^−7^ mol/s-m-Pa (used for Figure 7, Figure 8, Figure 9 and Figure 10 only)*Heat transfer factor f (Equation (7)):* 0.12 (used for Figure 7, Figure 8, Figure 9 and Figure 10 only) 


### 6.2. Extended PBR Simulations

The simulations for Figure 5 and Figure 6 assume the NH_3_ decomposition occurs in an adiabatic packed bed reactor external to any fuel cell. However, this decomposition can occur as part of the anode structure of the fuel cell [9]. In this case, H_2_ diffusion through the membrane for exothermic oxidation at the cathode accompanies the NH_3_ breakdown. In addition, heat transfer from the cathode across the membrane to the anode can supply some of the decomposition endothermicity. Additional PBR calculations were performed to crudely simulate these cases with only the engineering model (Equations (13)–(17)) and simplified rate expression (Equations (6) and (12)). The *Polymath*^®^ ODE solver [19] was used.

Figure 7, Figure 8, Figure 9 and Figure 10 show a sequence of additional cases that are easily obtained with the engineering model and overall rate expression, with details provided in Table 3. The three cases shown are H_2_ diffusion off (k_c_ = 0) and adiabatic (f = 0), diffusion on (k_c_ = 5 × 10^−7^ mol/s-m-Pa) and adiabatic, and diffusion on with heat transfer on (f = 0.12). The models of H_2_ diffusion and heat transfer are crude, but they illustrate the utility of the engineering model with the overall rate expression.
Figure 7Impact of H_2_ diffusion and heat transfer on axial temperature in PBR; 950 K feed temperature; pure NH_3_ feed = 1 × 10^−4^ mol/s (see Table 3 for remaining conditions). Calculations based on engineering model with overall decomposition rate expression Equations (6) and (12).
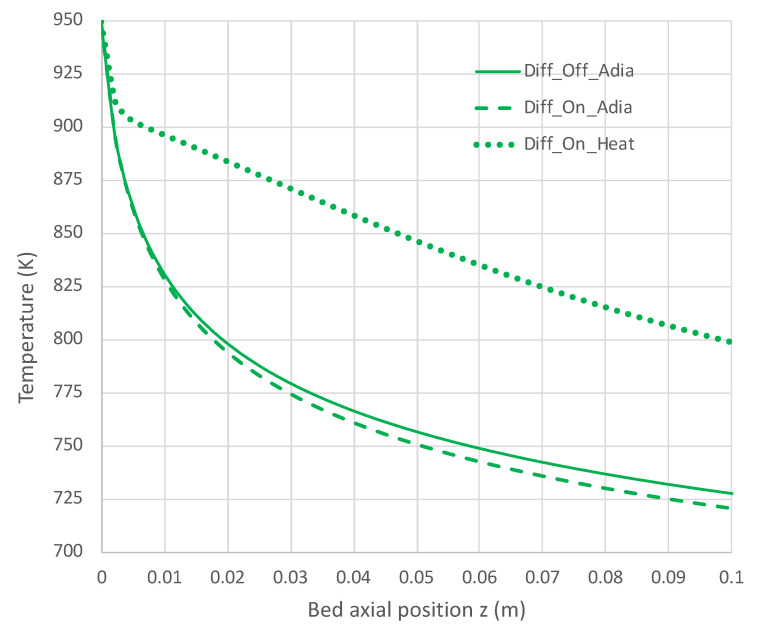



Under adiabatic conditions, the temperature drops rapidly due to the endothermicity of the NH_3_ pyrolysis, as shown in Figure 7. This is accompanied by only modest conversion (Figure 8). Loss of H_2_ by diffusion has little impact on the bulk flow temperature, even though the drop in H_2_ flow rate is large, as shown in Figure 10. Figure 9 shows that the NH_3_ flow rate has not dropped much, as is also evident in Figure 8, with only an approximately 23% conversion.

Allowing heat to flow into the packed bed has a significant impact. The slower temperature drop (Figure 7) results in a significantly smaller NH_3_ rate (Figure 9) and hence a larger NH_3_ conversion (Figure 8). Even more H_2_ is available for diffusion (Figure 10). The N_2_ flow rate also increases due to the higher NH_3_ conversion.

**Figure 8 molecules-28-06006-f008:**
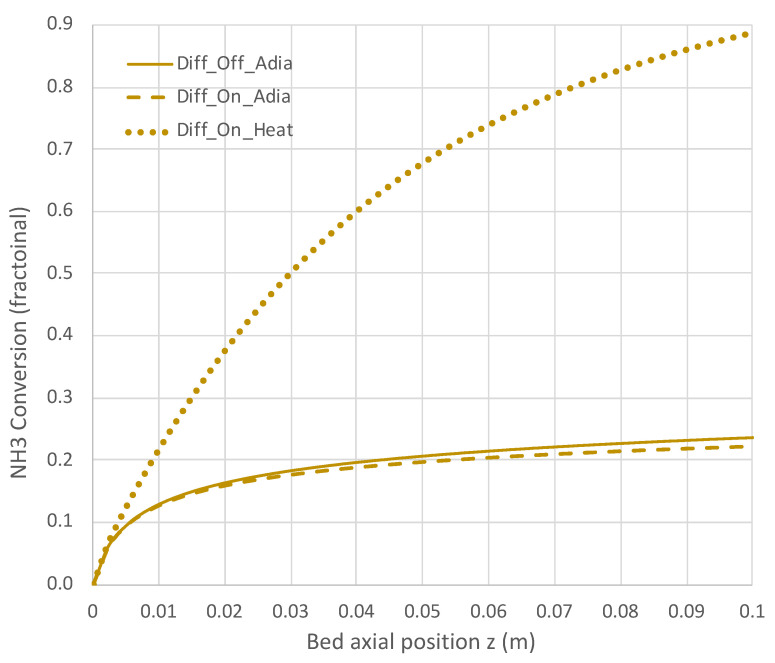
Impact of H_2_ diffusion and heat transfer on NH_3_ conversion in PBR; 950 K feed temperature; pure NH_3_ feed = 1 × 10^−4^ mol/s (see Table 3 for remaining conditions). Calculations are based on engineering model with overall decomposition rate expression (Equations (6) and (12)).

**Figure 9 molecules-28-06006-f009:**
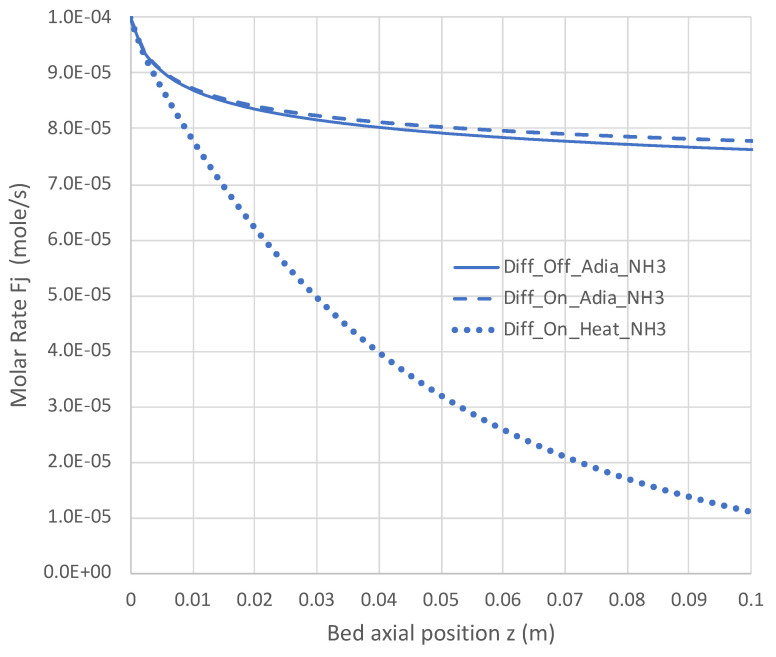
Impact of H_2_ diffusion and heat transfer on axial NH_3_ flow rate in PBR; 950 K feed temperature; pure NH_3_ feed = 1 × 10^−4^ mol/s (see Table 3 for remaining conditions). Calculations use engineering model with single overall decomposition rate expression (Equations (6) and (12)).

**Figure 10 molecules-28-06006-f010:**
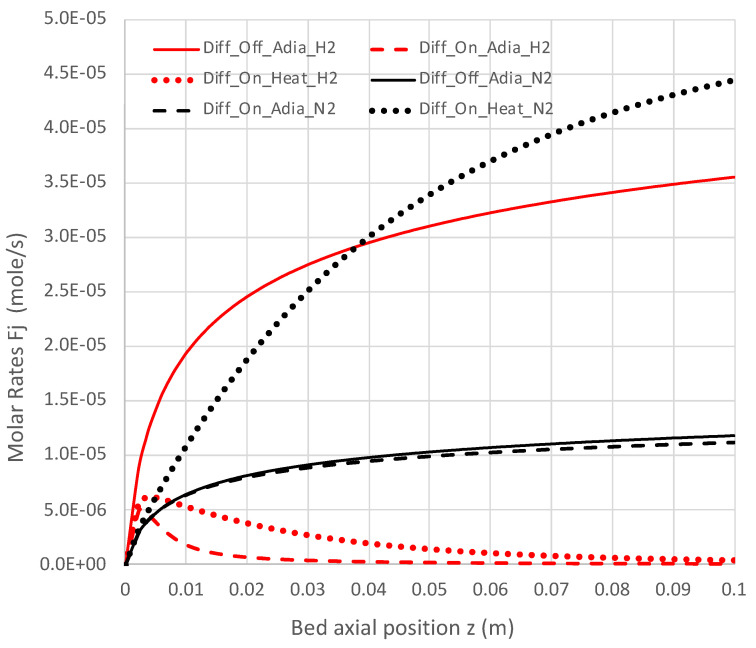
Impact of H_2_ diffusion and heat transfer on axial N_2_ and H_2_ flow rates in PBR; 950 K feed temperature; pure NH_3_ feed = 1 × 10^−4^ mol/s (see Table 3 for remaining conditions). Calculations use engineering model with overall decomposition rate expression (Equations (6) and (12)).

## 7. Conclusions

An overall rate expression, based on an assumed slow step (desorption of adsorbed N atoms to form N_2_) with five remaining elementary steps in fast pseudo-equilibria, has been derived to accurately simulate the decomposition of NH_3_ over a Ni-BaZrO_3_ catalyst at 1 × 10^5^ Pa at 650–950 K. The expression was calibrated using decomposition rates calculated with a six-step elementary mechanism from the literature proven independently against experimental data. The overall rate expression, with two temperature-dependent parameters, and its implementation in an engineering model successfully predicted NH_3_ decomposition performance in a packed bed reactor as calculated with the detailed mechanism. The utility of the engineering model with the overall rate expression was further demonstrated with simulations of NH_3_ decomposition in a packed bed reactor allowing for H_2_ diffusion and heat transfer in an approximation of a fuel cell anode feeding NH_3_. The methodology used in this work is independent of the choice of catalyst. It relies on the availability of a previously published and validated elementary reaction mechanism.

## Figures and Tables

**Figure 1 molecules-28-06006-f001:**
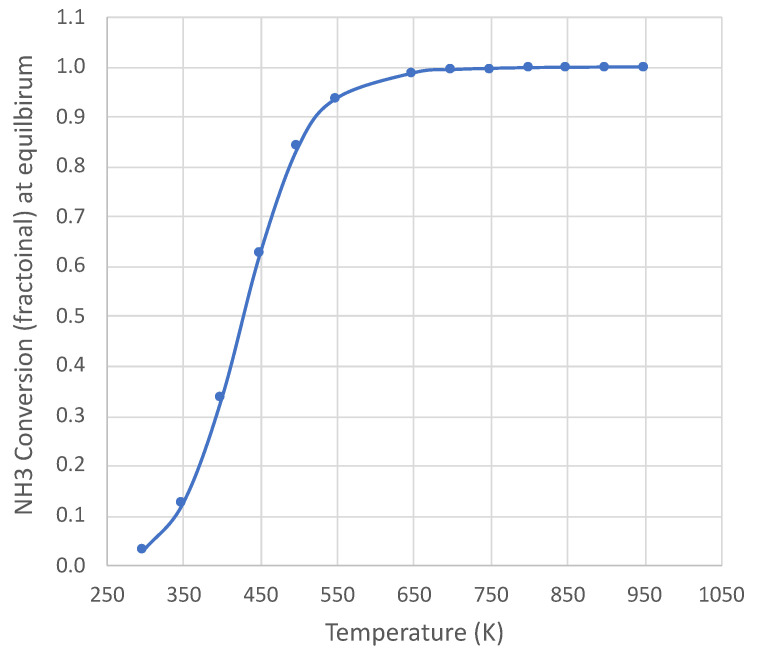
Equilibrium NH_3_ conversion for pure NH_3_ feed at 1 × 10^5^ Pa pressure. Calculation based on NASA CEA equilibrium code [10].

**Figure 2 molecules-28-06006-f002:**
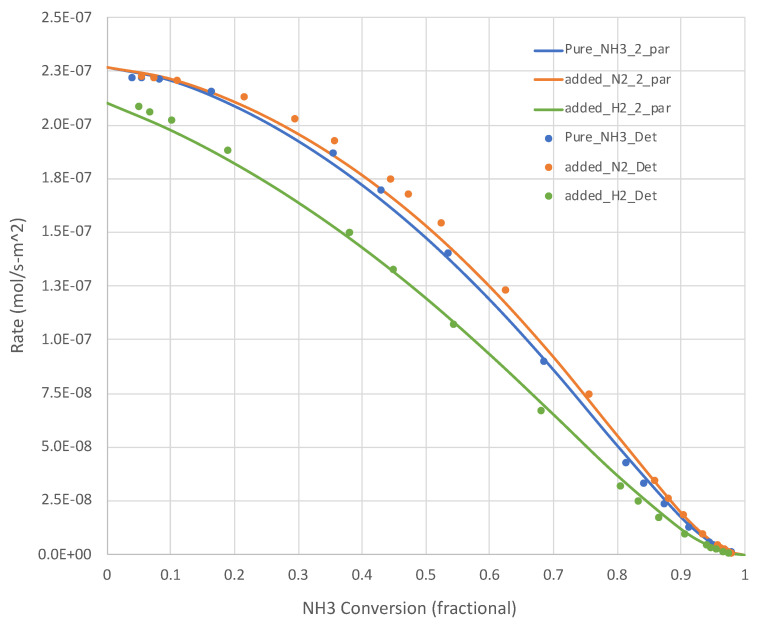
Calculated (*Detchem*^®^, Equation (6) overall rate expression with 2 parameters) NH_3_ decomposition rates at 650 K for both pure NH_3_ and mixed (25 mole% H_2_ or N_2_, bal. NH_3_) feeds (see Table 1). N H_3_ conversion varies as a function of total feed rate.

**Figure 3 molecules-28-06006-f003:**
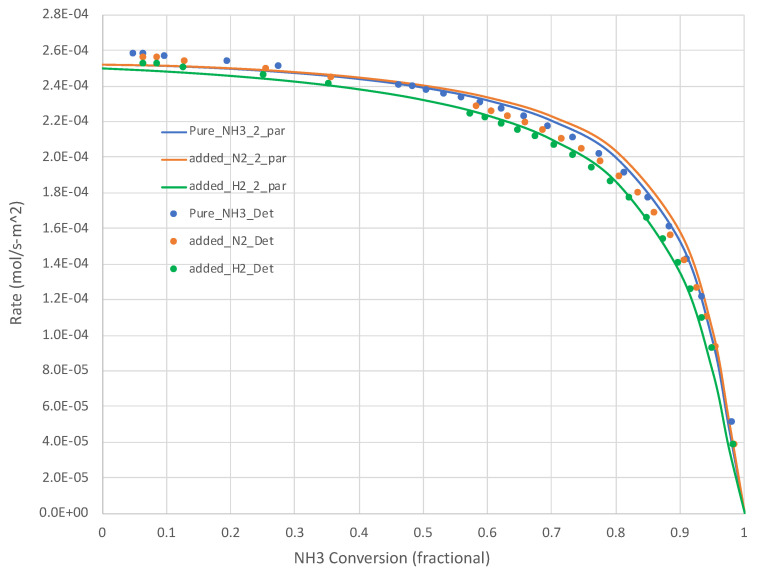
Calculated (*Detchem*^®^, Equation (6) overall rate expression with 2 parameters) NH_3_ decomposition rates at 950 K for both pure NH_3_ and mixed (25 mole% H_2_ or N_2_, bal. NH_3_) feeds (see Table 1). NH_3_ conversion varies as a function of total feed rate.

**Figure 4 molecules-28-06006-f004:**
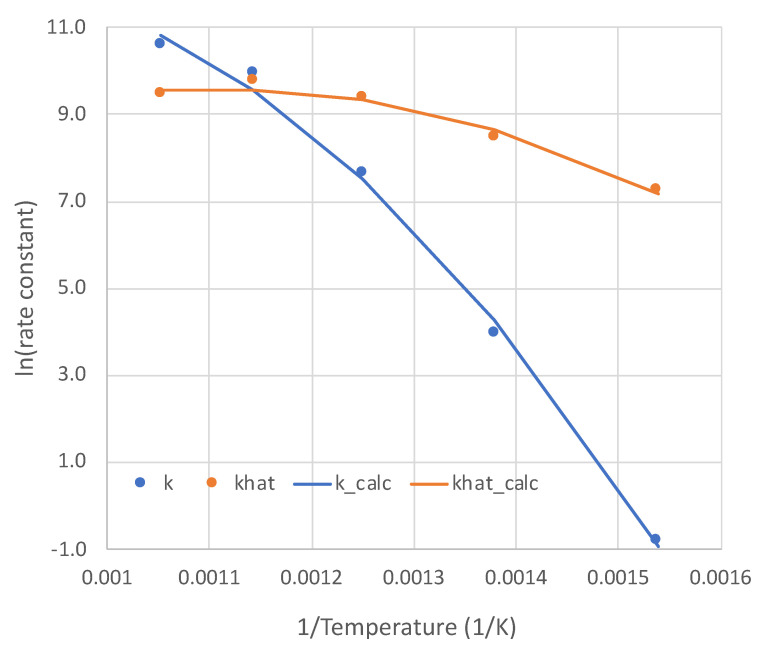
Temperature dependencies (point values and regressions) of the two parameters of overall kinetic rate expression (Equation (6)). Units: k (mol-Pa/s-m^2^); khat = $\hat{k}\;({\mathrm{Pa}}^{0.5})$.

**Figure 5 molecules-28-06006-f005:**
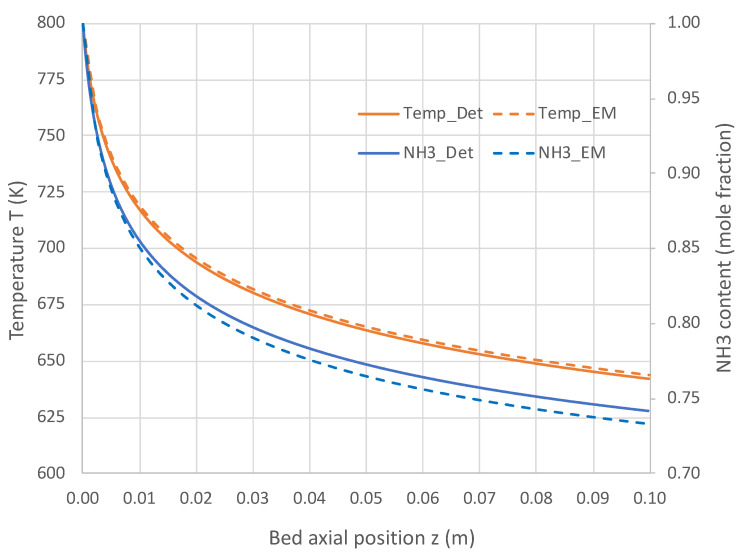
Comparison of engineering model (EM) with overall kinetic rate expression (Equation (6)) vs. *Detchem*^®^ with detailed mechanism for axial profiles in adiabatic packed bed reactor feeding pure NH_3_ at 1 × 10^−5^ mol/s and 800 K (see Table 3 for remaining conditions).

**Figure 6 molecules-28-06006-f006:**
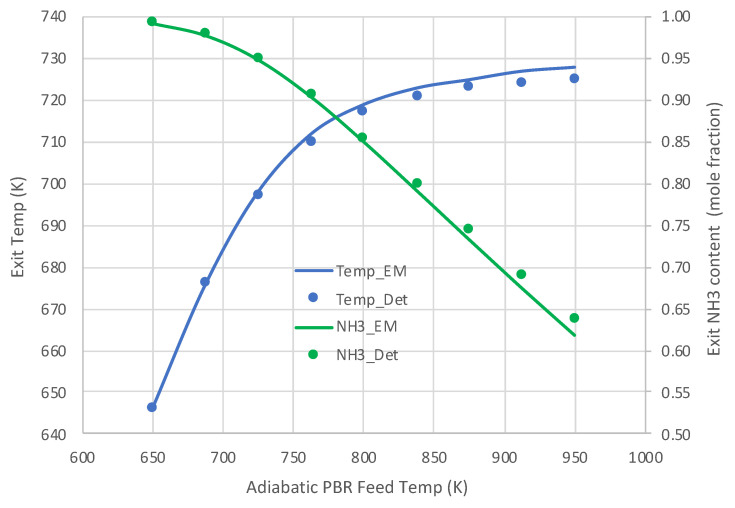
Comparison of engineering model (EM) with overall kinetic rate expression (Equations (6) and (12)) vs. *Detchem*^®^ with detailed mechanism for adiabatic PBR feeding pure NH_3_ at 1 × 10^−4^ mol/s (see Table 3 for remaining conditions).

**Table 1 molecules-28-06006-t001:** Data for *Detchem*^®^ CSTR^®^ Simulations.

	Feed Rate (mol/s)	
Temperature (K)	Lowest	Highest
650	1 × 10^−11^	1 × 10^−7^
725	1 × 10^−10^	1 × 10^−5^
800	1 × 10^−8^	1 × 10^−4^
875	1 × 10^−7^	1 × 10^−3^
950	1 × 10^−6^	1 × 10^−4^

*Pressure:* 1 × 10^5^ Pa (constant);  *Catalyst area:* 1.92 × 10^−2^ m^2^; *Reactor gas volume:* 3.33 × 10^−8^ m^3^; *Feed:* pure NH_3_, or 25 mol% H_2_ or N_2_ (balance NH_3_) at same total molar rate for selected cases.

**Table 2 molecules-28-06006-t002:** Rate constants for 2-parameter kinetic rate expression (Equation (6)). Form: $ln\left(k\;or\;\hat{k}\right)=a+b\left(1/T\right)+c{\left(1/T\right)}^{2}$ where T (K).

	For *k* (mol-Pa/s-m^2^)	$\mathrm{For}\;\hat{k}\;({\mathrm{Pa}}^{0.5})$
a	−5.996	−6.181
b	4.344 × 10^4^	2.849 × 10^4^
c	−2.610 × 10^7^	−1.287 × 10^7^

## Data Availability

Data is available from the author upon request.

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
