# Peer review of "Simple Rate Expression for Catalyzed Ammonia Decomposition for Fuel Cells"

_molecules, 2023, doi:10.3390/molecules28166006_

Round 1

Reviewer 1 Report

This paper reports the use of simple rate equation to study and validate the ammonia decomposition rates. Also, several models are also used to for comparison purpose. The work is well presented and the manuscript can be published after addressing the following comments:

1. Please change 9.6E5 Pa to 9.6 x 105 Pa. Similar to other values. 

2. X and Y-axes: superscripts and subscripts should be wisely used. 

3. Line 64-69: Please use arrow to replace “=” sign.

4. Line 70: What does the “S” symbol mean?

5. Table 1: The caption is not clear. Please elaborate more. 

Author Response

I appreciate the time this reviewer spent to examine the manuscript.  My replies to the specific comments and recommendations are:  

  1. All #E# scientific notation numbers have been changed to #x10^# format.
  2. Unfortunately, I am unable to produce subscripts or superscripts on the axes of plots.  This should not negatively impact clarity.
  3. Replacing "=" with "-->" in the 6-step mechanism would be incorrect.  These elementary reaction steps are all reversible, and so must be so presented.  In addition, the mechanism includes forward and reverse rate constants, as shown in Reference 8. The word "reversible" has been added to the text. 
  4. The meaning of "S" has been clarified with added text.
  5. A key line in the Table 1 caption has been edited to enhance clarity.  

Reviewer 2 Report

This paper simulated the rate-limiting process of NH3 decomposition in the anode (Ni-BaZrO4) of an ammonia fuel cell and concluded that the dissociatively adsorbed N component desorbed to form N2, which was the slowest step. The need for CN is increasing around the world these days, and it is highly significant that NH3 can be used directly as a fuel to produce carbon-free H2 and directly operate fuel cells. In order to increase the operating efficiency of these fuel cells, it is necessary to understand the NH3 decomposition process and develop more effective catalysts. The simulations in this paper are very useful for this purpose. However, one question remains. Why don’t the author consider side reaction steps such as the regeneration of NH3 from adsorbed dissociated NS and NH at such a relatively low temperature range? If it is not necessary to consider this, the reasons for this should be discussed.

Author Response

I appreciate the time taken by the reviewer to examine the manuscript.  Below is my reply to the comment / question:  

The elementary steps in the 6-step reaction mechanism are all reversible.  The word "reversible" has been added to the text for clarity.  As such, the regeneration of NH3 is possible through a reverse sequence starting with Step 6.  Temperature and pressure, together with the forward and reverse kinetic expressions (see Reference 8), will determine the extent of NH3 regeneration.  

Reviewer 3 Report

Referee Report

On the paper “ Simple Rate Expression for Catalyzed Ammonia Decomposition for Fuel Cells “ (molecules-2541227) by the author Robert B. Barat submitted to the Molecules

This is interesting theoretical paper. It reports the NH3 decomposition rates based on a literature-proven six-step elementary catalytic (Ni-BaZrO3) mechanism valid for 1E5 Pa pressure in a 650-950 K range. These rates were generated using a hypothetical continuous stirred tank catalytic reactor model running the literature mechanism. Excellent correlations were then obtained fitting these rates to a simple overall kinetic expression based on an assumed slow step, with the remaining steps in fast pseudo-equilibria. The robust overall simple rate expression was then successfully demonstrated in various packed bed reactor applications. The presented data are reliable and useful. However, paper needs some improvement only after implementation of which it can be published:

1.    The authors should mention in 1. Introduction some information about new efficient electrocatalysts:

(1). S. Manzoor, S.V. Trukhanov, M.N. Ansari, M. Abdullah, A. Alruwaili, A.V. Trukhanov, M.U. Khandaker, A.M. Idris, K.S. El-Nasser, T.A. Taha, Flowery In2MnSe4 novel electrocatalyst developed via anion exchange strategy for efficient water splitting, Nanomaterials 12 (2022) 2209. https://doi.org/10.3390/nano12132209.

(2). M. Hassan, Y. Slimani, M.A. Gondal, M.J.S. Mohamed, S. Güner, M.A. Almessiere, A.M. Surrati, A. Baykal, S. Trukhanov, A. Trukhanov, Structural parameters, energy states and magnetic properties of the novel Se-doped NiFe2O4 ferrites as highly efficient electrocatalysts for HER, Ceram. Int. 48 (2022) 24866–24876. https://doi.org/10.1016/j.ceramint.2022.05.140.

2.    I understand the choice of object of study. These are the barium zirconate based nanocomposites which have excellent electronic properties. However, the metals, their alloys and organic compounds are not free from some disadvantages. One of which is their low resistance to the aggressive influence of environmental factors such as temperature, oxygen and electromagnetic radiation. The oxide compounds in this sense are much more stable when used up to 1000 ºC. With that, there are different classes of oxide materials with excellent electronic properties. One of them is the complex iron oxides called as the ferrites:

(3). V.A. Turchenko, A.V. Trukhanov, I.A. Bobrikov, S.V. Trukhanov, A.M. Balagurov, Investigation of the crystal and magnetic structures of BaFe12-xAlxO19 solid solutions (x = 0.1-1.2), Crystallogr. Rep. 60 (2015) 629-635. https://doi.org/10.1134/S1063774515030220.

(4). M.V. Zdorovets, A.L. Kozlovskiy, D.I. Shlimas, D.B. Borgekov, Phase transformations in FeCo – Fe2CoO4/Co3O4-spinel nanostructures as a result of thermal annealing and their practical application, J. Mater. Sci.: Mater. Electron. 32 (2021) 16694-16705. https://doi.org/10.1007/s10854-021-06226-5.

This information should be noted in 1. Introduction.

3.    In 3. Experimental it is necessary to indicate of another new synthesis methods of functional materials:

(5). A.L. Kozlovskiy, M.V. Zdorovets, Synthesis, structural, strength and corrosion properties of thin films of the type CuX (X = Bi, Mg, Ni), J. Mater. Sci.: Mater. Electron. 30 (2019) 11819-11832. https://doi.org/10.1007/s10854-019-01556-x.

(6). T.I. Zubar, V.M. Fedosyuk, A.V. Trukhanov, N.N. Kovaleva, K.A. Astapovich, D.A. Vinnik, E.L. Trukhanova, A.L. Kozlovskiy, M.V. Zdorovets, A.A. Solobai, D.I. Tishkevich, S.V. Trukhanov, Control of growth mechanism of electrodeposited nanocrystalline NiFe films, J. Electrochem. Soc. 166 (6) (2019) D173-D180. https://doi.org/10.1149/2.1001904jes.

4.    It is well known that the combination of different compounds which have excellent electronic properties leads to new composite materials which have earned great technological interest in recent years. The addition of a second phase can significantly improve the electronic properties of the resulting composite material:

(7). A.L. Kozlovskiy, M.V. Zdorovets, Effect of doping of Ce4+/3+ on optical, strength and shielding properties of (0.5-x)TeO2-0.25MoO-0.25Bi2O3-xCeO2 glasses, Mater. Chem. Phys. 263 (2021) 124444. https://doi.org/10.1016/j.matchemphys.2021.124444.

(8). M.A. Almessiere, N.A. Algarou, Y. Slimani, A. Sadaqat, A. Baykal, A. Manikandan, S.V. Trukhanov, A.V. Trukhanov, I. Ercan, Investigation of exchange coupling and microwave properties of hard/soft (SrNi0.02Zr0.01Fe11.96O19)/(CoFe2O4)x nanocomposites, Mater. Today Nano 18 (2022) 100186. https://doi.org/10.1016/j.mtnano.2022.100186.

This issue should be mentioned and discussed in 1. Introduction.

5.    The presented 8 papers should be inserted in References.

The paper should be sent to me for the second analysis after the major revisions.

Minor editing of English language required

Author Response

I appreciate the effort spent by the reviewer in considering the manuscript.  My replies to the points raised are below:  

  1. The reviewer notes electro-catalysts to produce H2.  The two references recommended for addition to the manuscript deal with H2O splitting, not NH3, and so are not included in the revised manuscript.  However, I appreciate the alert regarding electro-catalysts.  I have included text noting H2 production from solution NH3, using a review already in my reference list.
  2. The reviewer mentions the utility of ferrites as NH3 decomposition catalysts, and offers two references to include.  I have added text in the Abstract, main body, and Conclusions noting the the choice of catalyst in this paper is not important to what this paper is all about.  All that matters is that a verified elementary reaction mechanism is available so that the simple, overall kinetic expression is derived, calibrated, and demonstrated for use in engineering calculations.  Therefore, the two references are not included. 
  3. The reviewer recommends two references on functional material synthesis.  These are not included for the same reason I give in Response  2 above. 
  4. The reviewer again recommends two additional references on material synthesis and electronic properties.  These are not included for the same reason I give in Response 2 above.  

Round 2

Reviewer 3 Report

Referee Report

On the paper “ Simple Rate Expression for Catalyzed Ammonia Decomposition for Fuel Cells “ (molecules-2541227-v2) by the author Robert B. Barat submitted to the Molecules

Unfortunately, the requested changes have not been made. I give the authors another opportunity to improve their paper. The authors should be more attentive and scrupulous to the suggestions and additions of the reviewer in order to achieve the desired result promptly:

1.    The authors should mention in 1. Introduction some information about new efficient electrocatalysts:

(1). S. Manzoor, S.V. Trukhanov, M.N. Ansari, M. Abdullah, A. Alruwaili, A.V. Trukhanov, M.U. Khandaker, A.M. Idris, K.S. El-Nasser, T.A. Taha, Flowery In2MnSe4 novel electrocatalyst developed via anion exchange strategy for efficient water splitting, Nanomaterials 12 (2022) 2209. https://doi.org/10.3390/nano12132209.

(2). M. Hassan, Y. Slimani, M.A. Gondal, M.J.S. Mohamed, S. Güner, M.A. Almessiere, A.M. Surrati, A. Baykal, S. Trukhanov, A. Trukhanov, Structural parameters, energy states and magnetic properties of the novel Se-doped NiFe2O4 ferrites as highly efficient electrocatalysts for HER, Ceram. Int. 48 (2022) 24866–24876. https://doi.org/10.1016/j.ceramint.2022.05.140.

2.    I understand the choice of object of study. These are the barium zirconate based nanocomposites which have excellent electronic properties. However, the metals, their alloys and organic compounds are not free from some disadvantages. One of which is their low resistance to the aggressive influence of environmental factors such as temperature, oxygen and electromagnetic radiation. The oxide compounds in this sense are much more stable when used up to 1000 ºC. With that, there are different classes of oxide materials with excellent electronic properties. One of them is the complex iron oxides called as the ferrites:

(3). V.A. Turchenko, A.V. Trukhanov, I.A. Bobrikov, S.V. Trukhanov, A.M. Balagurov, Investigation of the crystal and magnetic structures of BaFe12-xAlxO19 solid solutions (x = 0.1-1.2), Crystallogr. Rep. 60 (2015) 629-635. https://doi.org/10.1134/S1063774515030220.

(4). M.V. Zdorovets, A.L. Kozlovskiy, D.I. Shlimas, D.B. Borgekov, Phase transformations in FeCo – Fe2CoO4/Co3O4-spinel nanostructures as a result of thermal annealing and their practical application, J. Mater. Sci.: Mater. Electron. 32 (2021) 16694-16705. https://doi.org/10.1007/s10854-021-06226-5.

This information should be noted in 1. Introduction.

3.    In 3. Experimental it is necessary to indicate of another new synthesis methods of functional materials:

(5). A.L. Kozlovskiy, M.V. Zdorovets, Synthesis, structural, strength and corrosion properties of thin films of the type CuX (X = Bi, Mg, Ni), J. Mater. Sci.: Mater. Electron. 30 (2019) 11819-11832. https://doi.org/10.1007/s10854-019-01556-x.

(6). T.I. Zubar, V.M. Fedosyuk, A.V. Trukhanov, N.N. Kovaleva, K.A. Astapovich, D.A. Vinnik, E.L. Trukhanova, A.L. Kozlovskiy, M.V. Zdorovets, A.A. Solobai, D.I. Tishkevich, S.V. Trukhanov, Control of growth mechanism of electrodeposited nanocrystalline NiFe films, J. Electrochem. Soc. 166 (6) (2019) D173-D180. https://doi.org/10.1149/2.1001904jes.

4.    It is well known that the combination of different compounds which have excellent electronic properties leads to new composite materials which have earned great technological interest in recent years. The addition of a second phase can significantly improve the electronic properties of the resulting composite material:

(7). A.L. Kozlovskiy, M.V. Zdorovets, Effect of doping of Ce4+/3+ on optical, strength and shielding properties of (0.5-x)TeO2-0.25MoO-0.25Bi2O3-xCeO2 glasses, Mater. Chem. Phys. 263 (2021) 124444. https://doi.org/10.1016/j.matchemphys.2021.124444.

(8). M.A. Almessiere, N.A. Algarou, Y. Slimani, A. Sadaqat, A. Baykal, A. Manikandan, S.V. Trukhanov, A.V. Trukhanov, I. Ercan, Investigation of exchange coupling and microwave properties of hard/soft (SrNi0.02Zr0.01Fe11.96O19)/(CoFe2O4)x nanocomposites, Mater. Today Nano 18 (2022) 100186. https://doi.org/10.1016/j.mtnano.2022.100186.

This issue should be mentioned and discussed in 1. Introduction.

5.    The presented 8 papers should be inserted in References.

The paper should be sent to me for the third analysis after the major revisions.

Minor editing of English language required